# Mortality rate and associated factors among patients co-infected with drug resistant tuberculosis/HIV at Mulago National Referral Hospital, Uganda, a retrospective cohort study

**Joan Rokani Bayowa** [1]*, **Joan N. Kalyango**[1], **Joseph Baruch Baluku**[2], **Richard Katuramu**[3], **Emmanuel Ssendikwanawa**[1], **Jane Frances Zalwango**[1], **Rebecca Akunzirwe**[1], **Stella Maris Nanyonga**[1], **Judith Ssemasazi Amutuhaire** [1], **Ronald Kivumbi Muganga** [1], **Adolphus Cherop**[1]

**1** Clinical Epidemiology Unit, College of Health Sciences, Makerere University, Kampala, Uganda, **2** Mulago National Referral Hospital, Tuberculosis Unit, Kampala, Uganda, **3** Ministry of Health, Tuberculosis Control Program, Kampala, Uganda

* rokanijo@gmail.com

## Abstract

Drug resistant tuberculosis (DR-TB)/HIV co-infection remains a growing threat to public health and threatens global TB and HIV prevention and care programs. HIV is likely to worsen the outcomes of DR-TB and DR-TB is likely to worsen the outcomes of HIV despite the scale up of TB and HIV services and advances in treatment and diagnosis. This study determined the mortality rate and factors associated with mortality among persons on treatment co-infected with drug resistant TB and HIV at Mulago National Referral Hospital. We retrospectively reviewed data of 390 persons on treatment that had a DR-TB/HIV co-infection in Mulago National Referral Hospital from January 2014 to December 2019. Modified poisson regression with robust standard errors was used to determine relationships between the independent variables and the dependent variable (mortality) at bivariate and multivariate analysis. Of the 390 participants enrolled, 201(53.9%) were males with a mean age of 34.6 (±10.6) and 129 (33.2%,95% CI = 28.7–38.1%) died. Antiretroviral therapy (ART) initiation (aIRR 0.74, 95% CI = 0.69–0.79), having a body mass index (BMI)$\geq$18.5Kg/m$^2$ (aIRR 1.01, 95% CI = 1.03–1.17), having a documented client phone contact (aIRR 0.85, 95% CI = 0.76–0.97), having a mid-upper arm circumference,(MUAC) $\geq$18.5cm (aIRR 0.90, 95% CI = 0.82–0.99), being on first and second line ART regimen (aIRR 0.83, 95% CI = 0.77–0.89), having a known viral load (aIRR 1.09, 95% CI = 1.00–1.21) and having an adverse event during the course of treatment (aIRR 0.88, 95% CI = 0.83–0.93) were protective against mortality. There was a significantly high mortality rate due to DR-TB/HIV co-infection. These results suggest that initiation of all persons living with HIV/AIDS (PLWHA) with DR-TB on ART and frequent monitoring of adverse drug events highly reduces mortality.

**Data Availability Statement:** The data for replicating the findings from this study can be found at https://zenodo.org/record/7944671.

**Funding:** This research work was supported by Africa Center of Excellence in Materials Product Development and Nanotechnology (MAPRONANO ACE)-Makerere University that is supported by the World Bank.The funders had no role in study design, data collection and analysis, decision to publish, or preparation of the manuscript.

**Competing interests:** The authors have declared that no competing interests exist.

## Background

Globally, in 2020, 4% of the people were newly diagnosed with Multi-drug resistant(MDR-TB) or rifampicin resistant (RR-TB), the most effective first line drug, with 21% having reoccurrences of multidrug-resistant TB (MDR-TB) or RR-TB after being previously treated [1]. About 15% of adults with multidrug-resistant tuberculosis die during treatment, 21% have unknown outcomes, and only 56% complete treatment or are cured. People living with HIV/AIDS(PLWHA), who comprise around 9% of all persons on treatment with Tuberculosis globally [1].There were about 1.3 million deaths among HIV- negative people, and an additional 300 000 deaths from TB among people living with HIV/AIDA were reported [2].Uganda is one of the African countries marked by WHO as TB burdened country [3].

DR-TB remains a growing threat to public health and threatens global TB and HIV prevention and care [2] and this is much worse in resource limited and low-income countries due to inadequate availability of prompt diagnostic and treatment measures. In Uganda, the DR-TB epidemic has been driven by the HIV co- infection due to immune compromised system. Additionally, the risk of TB is higher among people living with HIV/AIDS than HIV negative patients [4–6].There is a dual potential of death due to both disease conditions. There is high mortality associated with DR-TB [7] but also mortality associated with HIV [2] due to opportunistic infections and drug related adverse reactions. HIV is also likely to worsen outcomes from DR-TB [8] and DR-TB is likely to worsen the outcomes from HIV [9]. Also, the medicines for both conditions are associated with adverse events and toxicities such as: hepatotoxicity, renal toxicity [10] which have a high potential to result in death. In addition, toxicities may at times result into drug withdrawal by clinicians and affect person on treatment drug adherence in turn affecting treatment outcomes. Additionally, second line TB drugs e.g. linezolid when combined with ART may increase immunosuppression, drug- drug interactions and lead to higher rates of toxicity, higher pill burden and greater non-compliance further worsening treatment outcomes [11, 12]. If these problems are not managed early, they may lead to catastrophic costs and increased morbidity and mortality. There is paucity of data in the Ugandan context as to whether the DR-TB/HIV places a person at increased risk for mortality. Whereas studies have evaluated predictors of mortality in DR-TB and have found HIV to be one of them; few have evaluated predictors of mortality among those with DR-TB/HIV co-infection specifically [13, 14]. More still, studies about DR-TB/HIV mortality have not been carried out in the Ugandan setting, thus data is highly needed in Uganda to determine the mortality rate and factors associated with mortality among persons on treatment with DR—TB/HIV co-infection so as to detect gaps in diagnosis, detection and monitoring of DR-TB to identify implementation challenges and to act as a basis for future research to close the gaps and reduce mortality from DR-TB/HIV co-infection in Uganda. Findings from this study could generate knowledge to improve clinical care and management of persons on treatment co-infected with DR-TB and HIV in Uganda as well as guide policy changes so as to improve survival among persons on treatment co-infected with DR-TB/HIV. The objectives of this study were to determine the mortality rate and factors associated with mortality among persons receiving healthcare with DR-TB/HIV co-infection at Mulago National Referral Hospital in Uganda.

## Methods

### Study design

This was a retrospective cohort that reviewed medical records of DR-TB/HIV persons on treatment who were registered at Mulago Tuberculosis unit from 1st January 2014 to 31st December 2019.

### Study setting and study population

The study was carried out at Mulago Hospital tuberculosis unit. Mulago Hospital is a national referral and teaching Hospital. The hospital serves a vast majority of all referred DR-TB persons on treatment in Uganda who are treated as in and out clients. The study population comprised of all persons on treatment with laboratory confirmed DR-TBand HIV positive results, having a treatment outcome recorded and having attended the Mulago tuberculosis Unit between 1st January 2014 and 31st December 2019.

### Sample size

A total of 412 records of DR-TB/HIV co-infected persons on treatment from the DR-TB register and client files were reviewed over a six-year period. The following eligibility criteria were applied to ensure that the study sample was correctly selected (Fig 1). The inclusion criteria included persons on treatment with a confirmed DR-TB and HIV positive results with a recorded outcome and had attended Mulago TB unit between January 2014 to December 2019.Clients s with incomplete records/missing data on age, sex, BMI and treatment start date were excluded.

### Data collection

A data abstraction tool was used to collect persons on treatment sociodemographics and corresponding clinical informationby searching through the DR-TB register and corresponding client files. Data was entered in EpiDATA version 4.00 and thereafter exported to an excel spreadsheet. Data was checked for completeness and correctness and inconsistencies were verified from case records and errors were corrected.

### Statistical analysis

Data were transferred from the excel spread sheets into STATA version 14.0 for analysis and missing data were checked for each variable. At univariate level, the data was presented as frequencies and percentages for categorical variables, medians with corresponding interquartile ranges and means and standard deviation for continuous variables.

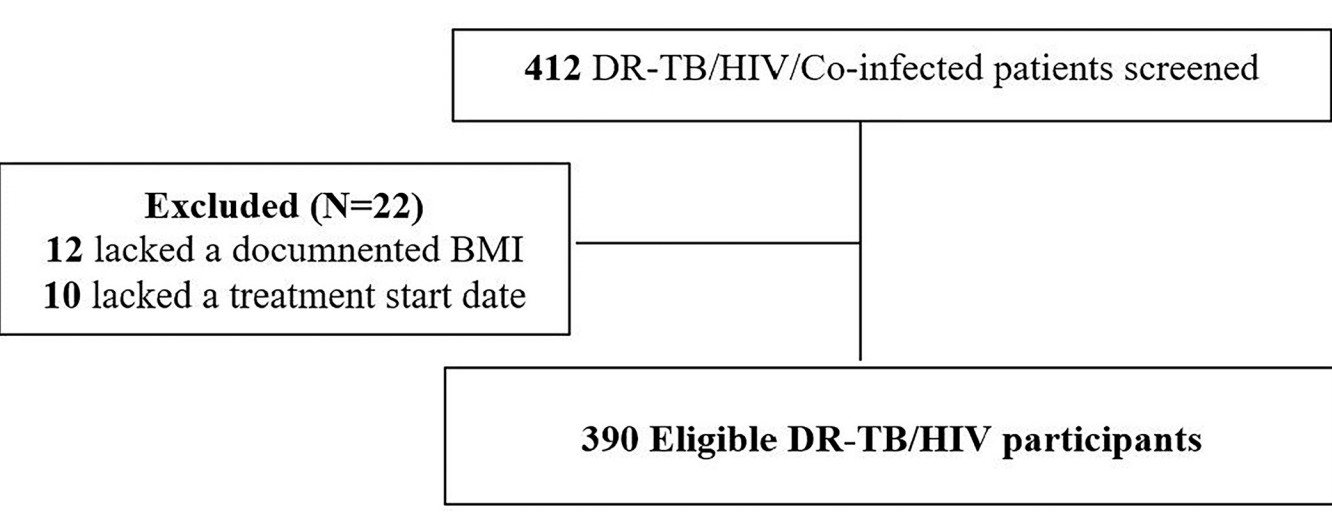

**Fig 1. Exclusion and inclusion criteria of the study cohort.**

At bivariate, modified poisson regression with robust standard errors with the family poisson and link log was used to determine relationships between the independent variables and the dependent variable (mortality). Independent variables with a pvalue <0.2 were taken to multivariate level for analysis. Incidence rate ratios (IRR) were the measure of association used.

At multivariate level, modified poisson regression with robust standard errors was used to determine the association between mortality and the independent variables. Independent variables with a pvalue <0.05 were the factors associated with mortality among DR-TB/HIV co-infected persons on treatment and were in the model. Interaction was assessed and potential confounders were adjusted for in the final multivariable models.

Incidence rate ratios were calculated with corresponding 95%confidence interval and alpha level of <0.05.

### Ethical considerations

The study was approved by Makerere University, School of Medicine Research and Ethics Committee and Mulago Research and ethics committee. Ethical and administrative approval numbers were (#REC REF 2020–063) and (MHREC 1832) respectively. All research assistants were trained on procedures to maintain confidentiality.

## Results

### Baseline characteristics

A total of 412 DR-TB/HIV persons on treatment were admitted from 1st January, 2014 to 31st December, 2019. Of these, 390 persons on treatment had a DR-TB confirmed and an HIV positive status recorded, out of which 22 persons on treatment were excluded from the study. Of the 22 individuals excluded, 12 lacked a documented body mass index (BMI) and 10 lacked a treatment start date as shown in (Fig 1). Therefore, 390 persons on treatment met the eligibility criteria (Fig 1). The participants aged between 1 to 80 years with a mean (SD) age at enrolment of 34.6 (± 10.5) years and majority were males (53.9%). Most of the participants had a documented phone contact (93.1%), no history of TB treatment (51.8%), had pulmonary Tuberculosis (97.1%) and a bacteriologically positive culture result at baseline (57.3%). Additionally, most of the participants had started ART (95.5%) and were on first line regimen (62.1%) with majority having an unknown viral load (75.9%). Most of the participants had had an adverse event (62.8%), a BMI $\geq$18.5Kg/m$^2$ (63.3%) and a mid-upper arm circumference (MUAC) $\geq$ 18.5 cm (95.6%) in (Table 1).

### Mortality rate

The mortality rate among persons on treatment co-infected with DR-TB/HIV at Mulago Hospital was 33.2% (95% CI: 28.7–38.1) with majority of the participants having a median time to mortality of $\leq$ 2.2 months from the time of treatment initiation. Most deaths were recorded in 2016 (35 deaths) as shown in Fig 2.

### Factors associated with mortality among persons on treatment co-infected with DR-TB/HIV at bivariate analysis

At bivariate analysis, having a phone contact (IRR = 0.86 95% CI: 0.75–0.98, p-value = 0.024), starting antiretroviral therapy (IRR = 0.62 95% CI:0.56–0.69, p-value = <0.001), mid upper arm circumference of $\geq$18.5cm (IRR = 0.88 95% CI:0.80–0.96, p- value = 0.004) a body mass index of$\geq$18.5kg/m$^2$ (IRR = 0.88 95% CI:0.82–0.94, p- value = <0.001), being on third line ART regimen (IRR = 0.76 95% CI:0.71–0.81, p-value = <0.001) and experiencing an adverse

**Table 1. Sociodemographic and clinical characteristics of the study participants (N = 390).**

| Characteristic | Frequency (%) |
| --- | --- |
| **Sex** | |
| Male | 210(53.85) |
| Female | 180(46.15) |
| **Age** | |
| < 35 | 201(51.54) |
| ≥ 35 | 189(48.46) |
| **BMI Catergory** | |
| < 18.5kg/m2 | 143(36.67) |
| ≥18.5Kg/m2 | 247(63.33) |
| **Case** | |
| No History of TB Rx | 202(51.79) |
| History of TB Rx | 188(48.21) |
| **Phone** | |
| No | 27(6.92) |
| Yes | 363(93.08) |
| **MUAC Catergory** | |
| < 18.5cm | 17(4.36) |
| ≥18.5cm | 373(95.64) |
| **Site of TB(n = 383)** | |
| Pulmonary TB(PTB) | 372(97.13) |
| Extrapulmonary TB(EPTB) | 2(0.52) |
| Both PTB &EPTP | 6(1.57) |
| Unspecified | 3(0.78) |
| **Bacteriological culture at baseline(n = 281)** | |
| Negative | 120(42.7) |
| Positive | 161(57.3) |
| **Adverse events** | |
| No | 145(37.2) |
| Yes | 245(62.8) |
| **ART Started(n = 382)** | |
| No | 17(45) |
| Yes | 365(95.5) |
| **ART Regimen** | |
| First line | 242(62.05) |
| Second line | 18(4.36) |
| Third line | 130(33.38) |
| **Viral load (n = 381)** | |
| Detectable | 37(9.7) |
| undetectable | 55(14.4) |
| unknown | 289(75.9) |

BMI-Body mass index, MUAC-Mid-upper arm circumference, ART-Antiretroviral Therapy TB-Tuberculosis-Treatment

event (IRR = 0.81 95% CI: 0.76–0.87, p-value = <0.001) were found to be protective against mortality. However, having an unspecified site of TB location (IRR = 1.91 95% CI: 1.16–1.23, p-value = <0.001) and having an unknown viral load (IRR = 1.91 95% CI: 1.08–1.31, p-value = <0.001) increased the likelihood of death as shown in (Table 2) below.

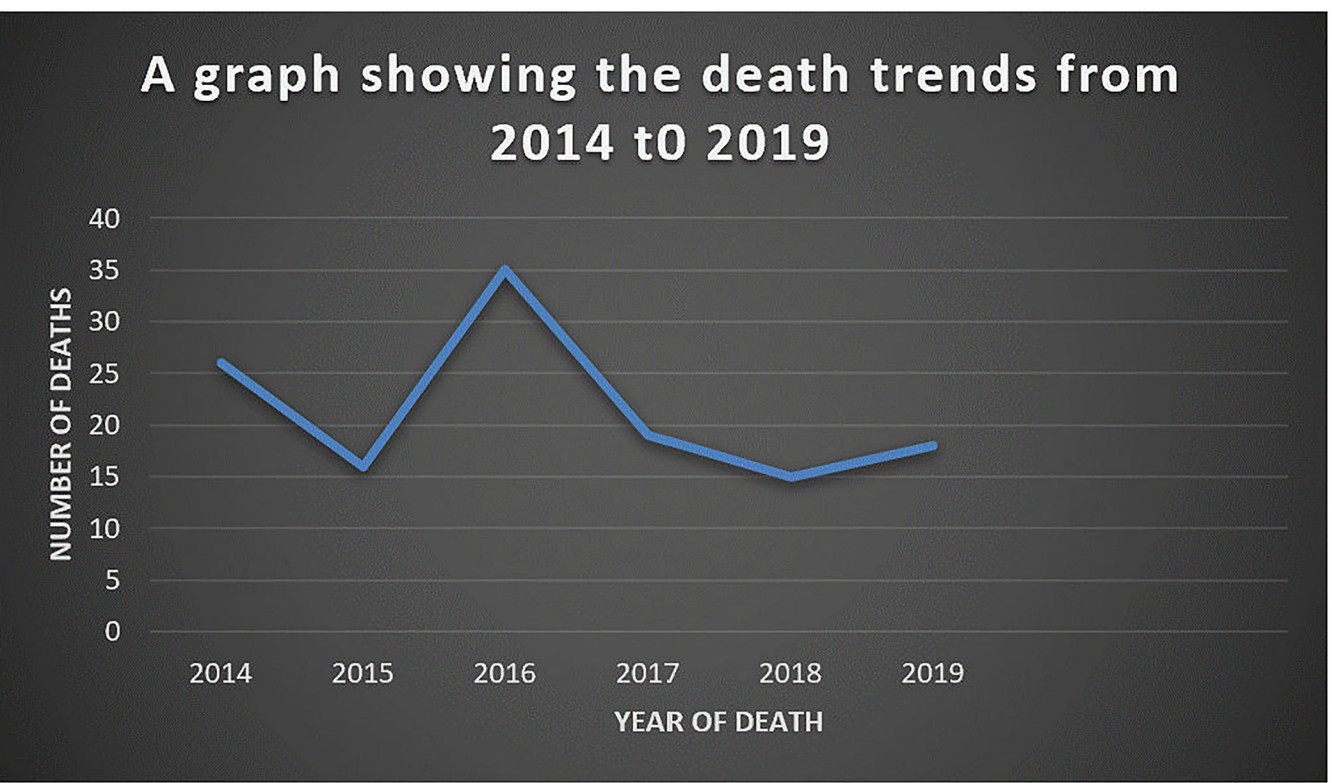

**Fig 2. A graph showing the death trends among persons on treatment co-infected with DR-TB/HIV in Mulago National Referral Hospital.**

## Factors associated with mortality among persons on treatment co-infected with DR-TB/HIV at multivariate analysis

Persons on treatment who started ART as they were taking their DR-TB treatment had a 25.7% reduced likelihood of death as compared to those that hadn't started ART (IRR = 0.74 95% CI:0.69–0.80, p-value = <0.001). Compared with normal BMI, those who had a BMI<18.5kg/m$^2$ (aIRR = 0.91 95% CI: 0.85–0.97, pvalue = 0.002) were more likely to die. Persons on treatment with an adverse event had 12.2% reduction in the risk of death as compared to those that didn't have an adverse event (aIRR = 0.88 95% CI: 0.81–0.93, p- value = <0.001). Additionally, those that had a MUAC of <18.5 cm were more likely to die as compared to those with a normal MUAC (aIRR = 0.90 95% CI:0.82–0.99, p-value = 0.004). Compared with persons on treatment that were on first and second line ART regimen, those on third line regimen were more likely to die (aIRR = 0.83 95% CI:0.77–0.89,p-value = <0.001) whereas persons on treatment with unknown viral load were more likely to die as compared to those that had a detectable or undetectable viral load (IRR = 1.09 95% CI:1.00–1.21, p-value = 0.049), as shown in (Table 2).

Persons on treatment on Levofloxacin had a 16% reduced likelihood of death compared to their counterparts that were not (aIRR = 0.84 95% CI:0.74–0.95, p-value = 0.007). Those on Moxifloxacin had a 22% reduction in the risk of death compared to those on other drugs (aIRR = 0.78 95% CI:0.67–0.89, p-value = 0.001) and those on Bedaquiline had a 9% reduction in the risk of death compared to those that were on other TB medications (aIRR = 0.91 95% CI:0.85–0.97, p-value = 0.004) as shown in (Table 3).

**Table 2. Factors associated with mortality among clients co-infected with DR-TB/HIV at Mulago Hospital, Uganda.**

| Characteristic | | Bivariate | | | | Multivariate | |
|---|---|---|---|---|---|---|---|
| | | Died | | IRR (95% CI) | Pvalue | aIRR(95%CI) | Pvalue |
| | | Yes (n%) | No (n %) | | | | |
| Sex | Male | 56(14.43) | 122(31.44) | | | | |
| | Female | 73(18.81) | 137(35.31) | 0.98(0.93-1.04) | 0.491 | | |
| Age | < 35 | 72(18.56) | 129(33.25) | | | | |
| | ≥ 35 | 57(14.69) | 130(33.51) | 1.03(0.98-1.09) | 0.264 | 1.02(0.98-1.07) | 0.358 |
| BMI at baseline | < 18.5kg/m2 | 63(44.1) | 80(55.1) | | | | |
| | ≥18.5Kg/m2 | 65(26.32) | 182(73.68) | 0.88(0.82-0.94) | 0.000* | 0.91(0.85-0.97) | 0.007 |
| Case | No History of TB Rx | 65(16.75) | 135(34.79) | | | | |
| | History of TB Rx | 64(16.49) | 124(31.96) | 0.99(0.94-1.05) | 0.748 | | |
| Phone (YES) | | 114(29.38) | 247(63.66) | 0.86(0.75-0.98) | 0.024 | | |
| MUAC baseline | < 18.5cm | 2(0.52) | 15(3.87) | | | | |
| | ≥ 18.5cm | 127(32.73) | 244(62.89) | 0.88(0.81-0.96) | 0.004 | 0.90(0.82-0.99) | 0.040* |
| Site of TB(n=383) | Pulmonary TB | 123(32.11) | 249(65.01) | | | | |
| | Extrapulmonary TB | 1(0.26) | 1(0.26) | 0.89(0.57-1.43) | 0.651 | | |
| | Both PTB &EPTP | 2(0.52) | 4(1.04) | 0.99(0.79-1.26) | 0.989 | | |
| | Unspecified | 0(0.00) | 3(0.79) | 1.19(1.16-1.23) | 0.000* | | |
| Sputum culture at baseline (n=281) | Negative | 24(8.21) | 96(34.29) | | | | |
| | Positive | 40(14.29) | 121(43.21) | 1.03(0.98-1.08) | 0.268 | | |
| Adverse events | No | 78(20.10) | 65(16.75) | | | | |
| | Yes | 51(13.14) | 194(50.0) | 0.81(0.76-0.87) | 0.000* | 0.88(0.83-0.93) | 0.000* |
| ART Started(n=382) | No | 16(4.19) | 1(0.26) | | | | |
| | Yes | 111(29.06) | 254(66.49) | 0.62(0.56-0.69) | 0.000* | 0.74(0.69-0.79) | 0.000* |
| ART Regimen | First line | 46(11.86) | 196(50.52) | | | | |
| | Second line | 3(0.77) | 15(3.87) | 1.10(0.92-1.12) | 0.797 | 0.99(0.92-1.09) | 0.937 |
| | Third line | 80(20.62) | 48(12.37) | 0.76(0.71-0.81) | 0.000* | 0.83(0.77-0.89) | 0.000* |
| Viral load (n=381) | Detectable | 13(3.42) | 24(6.32) | | | | |
| | undetectable | 2(0.53) | 52(13.68) | 0.98(0.89-1.09) | 0.756 | 1.03(0.94-1.13) | 0.572 |
| | unknown | 109(28.68) | 180(47.37) | 1.19(1.08-1.31) | 0.000* | 1.09(1.00-1.21) | 0.049 |

## Discussion

The study aimed at ascertaining the mortality rate and factors associated with mortality among DR-TB/ HIV co-infected persons on treatment at Mulago Hospital. The epidemic levels of drug resistant tuberculosis/HIV co- infection are growing threat to public health and threaten global TB and HIV prevention and care because they are associated with high mortality rates [2].This study reported a 33.2% mortality among DR-TB/HIV co- infected persons on treatment. This implied that approximately 33 persons receiving healthcare died in every 100 clients that had the co- infection. This is consistent with a Ugandan study by Walusimbi et al, that registered high deaths in 2016 [15].This trend could be explained by provider and health system barriers, including lack of knowledge, skills and negative attitude on TB among health providers, stockout of supplies such as medicines, high staff turnover, inability to track and follow-up TB persons receiving healthcare, and poor service coordination [16].Additionally, comorbidities with TB have also been found to a major risk for TB mortality [17, 18].

However, this was much higher than that reported by the East African Community (EAC) with Uganda inclusive that reported 24% TB deaths [19].This high mortality threatens Uganda's progress against TB and HIV fight towards her goals of zero deaths from DR-TB/HIV

**Table 3. Showing the Group A second line TB drugs at bivariate and multivariate analysis.**

| DR-TB drugs | Frequency (n %) | Died | | | | | |
|---|---|---|---|---|---|---|---|
| Group A | | Yes (n%) | No(n%) | IRR (95% CI) | Pvalue | aIRR (95% CI) | Pvalue |
| **Levofloxacin (n = 387)** | | | | | | | |
| No | 300(77.5) | 98(25.3) | 202(52.5) | | | | |
| Yes | 87(22.5) | 31(8.0) | 56(14.5) | 0.98(0.92–1.05) | 0.612 | 0.84(0.74–0.95) | 0.007 |
| **Moxifloxacin(n = 387)** | | | | | | | |
| No | 331(85.5) | 119(30.8) | 212(54.8) | | | | |
| Yes | 56(14.5) | 10(2.6) | 46(11.9) | 0.90(0.85–0.96) | 0.001 | 0.78(0.67–0.89) | 0.001 |
| **Bedaquiline(n = 387)** | | | | | | | |
| No | 338(87.3) | 122(31.5) | 216(55.8) | | | | |
| Yes | 49(12.7) | 7(1.8) | 42(10.9) | 0.88(0.83–0.94) | 0.000* | 0.91(0.85–0.97) | 0.004 |
| **Linezolid (n = 388)** | | | | | | | |
| No | 378(97.4) | 127(0.5) | 251(64.7) | | | | |
| Yes | 10(2.6) | 2(0.5) | 8(2.1) | 0.92(0.80–1.06) | 0.274 | 0.95(0.82–1.11) | 0.552 |

IRR- Incidence rate ratios, aIRR-adjusted Incidence rate ratios Pvalue-Probability Value, CI-Confidence Interval

persons on treatment as they are among the most vulnerable groups rapidly progressing to severe disease states or even death. This mortality rate is an overall marker of Uganda's health status as country and it affects economic development as young people are lost without reaching their years of full potential. It also exerts a lot of financial burden [20] to the country as a lot of funds have to be secured to buy drugs, testing kits etc. It further tells us about how interventions geared towards reduction of spread of DR-TB have been taken up by the population. These findings were consistent with those in Lesotho that reported a 34% mortality and 38% that was reported by Singh et al [2, 21]. However, these findings were different from a study done in South Africa that reported a 23.4% mortality [9], 19% in a multicenter cohort done in Abkhazia, Armenia, Colombia, Kenya, Kyrgystan, Swaziland and Uzbekistan [22],18% in a systematic review done in sub-Saharan Africa [23] and 11.4% among children [2].All these mortality rates were much lower than those reported in our study. The observed difference could be due to the differences in time, study population characteristics and DR-TB treatment practices/regimens in Uganda. Additionally, it could be due to improved compliance in documentation of persons receiving healthcare outcomes, different comprehensive models of DR-TB/HIV care and expertise through trainings as well continued efforts in implementation of drug resistant TB/HIV management policies in the country. Another study from South Africa reported a 42% mortality which was much higher than that that we found out in our study [24].This could be due to the fact that their study was done between 2001 to 2007 compared to ours that was done in 2019,and there had been a lot of changes in DR- TB policies, regimens and management.

## Factors associated with mortality

An adverse event was significantly associated with mortality. Having an adverse event was protective against mortality. There was a 12.2% reduced likelihood among persons on treatment that had an adverse event compared to those that didn't have. This finding wasn't surprising and was consistent with findings from a study done in India [25]. This could possibly be due to frequent monitoring of participants in our study than those in previous studies, thus increasing the chance of detecting adverse events. More still, persons on treatment who experienced adverse events were adherent to DR-TB therapy hence having a higher dose exposure,

and could have been monitored more closely by physicians, and thus were more likely to achieve favorable treatment outcomes. However these findings were different from another study that was done in South Africa [26] that found no significant association between having an adverse event and mortality, however there was infrequent documentation of specific drug associated events. This study found out that having a documented phone contact of the persons on treatment was protective against mortality. This is because presence of the phone contact would ease tracing and communication with the patient to ensure complete follow up, monitoring and ensuring proper adherence and compliance as well as tracing lost to follow up persons on treatment. It also ensured creation of a relationship and social support by the health care provider and the persons on treatment. These study findings were consistent with a study carried in Malawi [27]. The availability of phone contacts was an effective way of identifying outcomes of LTFU persons on treatment and following up persons receiving healthcare if they missed their drug and review appointment dates thus enhancing adherence in turn reducing death. Additionally, via telephone, persons on treatment were more likely to be located on a first attempt, admit leaving the clinic, and reveal their outcome status.

Having a BMI <18.5Kg/m$^2$ was a risk factor to mortality. There was a 9% reduced likelihood in mortality among persons on treatment who had a BMI ≥18.5kg/m2 than those who had a BMI<18.5kg/m2. These findings were consistent with those from different previous studies by [21, 28–30] in Mumbai, South Africa, Lesotho and Europe respectively. This is because HIV and TB are linked to malnutrition and wasting syndrome that cause severity of TB and HIV infection. Additionally, persons on DR-TB treatment experienced severe gastrointestinal intolerance (nausea, vomiting and gastritis) and drug toxicities during treatment that cause malnutrition and this may reduce the survival probability of the participants. This is scientifically supported since anti-TB drugs have serious adverse effects including nausea, vomiting and electrolyte disturbance which leads to poor prognosis. Additionally, BMI is a very important aspect in starting DR-TB therapy thus influences treatment initiation and persons receiving treatment, treatment outcomes.

Starting ART was significantly associated with mortality in this study. There was a 26% reduced likelihood of death among persons on treatment who had started ART as compared to those that hadn't started ART [31]. Our findings were consistent with those of Blanc et al that found out that the risk of death was significantly reduced in persons on treatment receiving ART earlier [32]. This is because by the time this study was undertaken, antiretroviral treatment (ART) was rolled out countrywide and was available in Uganda. This could also be due to th fact that starting ART reduces immunosuppression and opportunistic infections thus improving the health status [33]. Additionally, there has been increased voluntary HIV testing and counseling (VCT) at all entry points for persons receiving healthcare to receive ART, and social support has been given by clinicians due to stigma that was associated with HIV, most persons receiving healthcare are willing to be tested to know their HIV status. This is reflected by 100% of the clients in our study with a known HIV status. Indeed, the wide availability of ARVs in this period must have greatly reduced mortality levels of this cohort of DR-TB/HIV persons on treatment.

Mid Upper arm circumference (MUAC) of less than 18.5cm at baseline was a risk factor to mortality. There was a 9% reduced likelihood of death among persons on treatment that had a MUAC of ≥18.5cm as compared to those that had a MUAC of <18.5cm. This is because MAUC predicts the nutritional state of a person on treatment thus results into a very good stable environment for the person on treatment to take their medication well as they are being monitored by the clinician. Additionally, MUAC is a very important parameter while initiating a person on DR-TB drugs and affects treatment outcome before and after. These findings were consistent with a study done in Philippines that established that there was an association

between under-nutrition assessed using MUAC with risk of death [34]. Another possible explanation might be that DR-TB causes secondary malnutrition as persons on treatment lose appetite, nutrient, and micro-nutrient. This is also evidenced by fact that mal-absorption and altered metabolism resulted in poor survival. In turn, under nutrition could lead to secondary immunodeficiency which exacerbates poor survival of DR-TB persons on treatment [35].

Being on third line ART regimen reduced the risk of death by 17%. This is explained by the fact that after failure of both first line and second line ART regimens, a person is switched to third line and they are followed up by the clinicians to see their state of health leading to better treatment outcomes.

Persons on treatment that received Bedaquiline had a 9% reduction in risk of death compared to those that were on other DR-TB medication. This is consistent with results from a South African retrospective study and a meta-analysis [36] that found out the Bedaquiline had a reduced odds of death among persons on treatment with DR-TB and HIV infection [37, 38]. Persons on treatment that received Moxifloxacin and Levofloxacin had a 22% and 16% reduced likelihood of death as compared with their counterparts that were on different DR-TB medication. The findings in this study were consistent with findings from other studies that found that using at least of one WHO Group A drug, specifically use of moxifloxacin, levofloxacin, bedaquiline, or linezolid were associated with significantly decreased odds of death [36, 39].

Having an unknown and an undetectable viral load increased the likelihood of death by 19% and 3% respectively among the DR-TB/HIV co-infected persons on treatment as compared those that had a detectable viral load. This can be explained by the fact that those persons on treatment with a detectable viral load were frequently followed up by the health care personnel, tested and were encouraged to be adherent to their medications as compared to their counterparts that could have lost their adherent principles leading to an increase in viral load thus leading to poor outcomes [40, 41]. Additionally, persons on treatment with undetectable and unknown viral loads may be having other comorbidities and underlying illnesses that could be the risk factors of death other than DR-TB/HIV infection [18, 42–44].

## Strengths and limitations

This study depended on already collected data from medical records, hence may not have measured all the possible confounders. There was missing data on very important variables e.g., viral load that reduced the sample size and power. The study was prone to misclassification bias that could lead to biased estimates of the outcome. This study was conducted in a national referral hospital that could lead to referral bias thus limiting limits the generalizability of our findings to this referral hospitals. Mortality was confirmed in the hospital and for persons receiving healthcare who died at home, regular tracing and reminders for visits helped in ascertaining home-related DR-TB/HIV mortality. Certain variables like BMI, MUAC and sputum culture were collected at baseline to prevent different exposure duration thus preventing over or under-estimation of the effect size. Modified poisson Regression with robust standard errors was used and it provided unbiased estimates of IRRs. The existence of other comorbidities like diabetes was not established in this study and yet they are very important predictors of mortality.

## Conclusions

Having an adverse event, starting ART, body mass index $\geq$ 18.5 Kg/m$^2$, starting antiretroviral therapy, and mid upper arm circumference of $\geq$ 18.5 cm and being on third line ART regimen were protective against mortality among DR-TB/HIV co-infected persons on treatment. This study provided knowledge while also emphasizing what other researchers have found out to

clinicians and healthcare practitioners in clinical care and management of persons on treatment co-infected with DR-TB and HIV in Uganda.

## Recommendations

All persons on treatment phones should be documented for easy follow up and tracing as well as carrying out wellness visits. Ministry of Health and also other stake holders should encourage clinicians and all health care workers to initiate all DR-TB/HIV co-infected persons on antiretroviral therapy (ART) as well as monitor and manage adverse events as they lead to high treatment success.

Active case search finding, DR-TB/HIV mortality surveillance and practices on good antibiotic stewardship are recommended as they will help us substantially reduce the global burden of DR-TB. The Ugandan government should also increase funds for research in the DR-TB/HIV field to help develop molecular tests that are highly specific and sensitive in timely diagnosis of drug resistance.

DR-TB/HIV mortality surveillance should be heightened to ensure that no deaths are missed in order to design interventions that can curb the deaths.

Uganda, as one of the high burdened countries, should have political commitment and funding interms of DR-TB/HIV absolute numbers or severity and promote monitoring of progress.

Programs should be designed on incentivizing persons on treatment that adhere to their treatment and appointment dates all geared to eradicating antimicrobial resistance.

Accurate point-of-care tests based on whole genome sequencing should be done directly on sputum samples because such tests allow for rapid diagnosis and efficient individual based treatment of DR-TB.

## Acknowledgments

We thank all the people who were involved in the synthesis of this research up to its completion, and we are so grateful to Mulago Tuberculosis unit for the administrative support, willingly availing the data, hospital staff, and research assistants for their involvement in this study, as well as the staff and students in the Clinical Epidemiology Unit, College of Health Sciences, Makerere University.

## Author Contributions

**Conceptualization:** Joan Rokani Bayowa, Joan N. Kalyango, Joseph Baruch Baluku, Emmanuel Ssendikwanawa, Jane Frances Zalwango, Rebecca Akunzirwe, Stella Maris Nanyonga, Judith Ssemasazi Amutuhaire, Ronald Kivumbi Muganga, Adolphus Cherop.

**Data curation:** Joan Rokani Bayowa, Joan N. Kalyango, Emmanuel Ssendikwanawa, Jane Frances Zalwango, Rebecca Akunzirwe, Stella Maris Nanyonga, Judith Ssemasazi Amutuhaire, Ronald Kivumbi Muganga, Adolphus Cherop.

**Formal analysis:** Joan Rokani Bayowa, Joan N. Kalyango, Emmanuel Ssendikwanawa, Jane Frances Zalwango, Rebecca Akunzirwe, Stella Maris Nanyonga, Judith Ssemasazi Amutuhaire, Ronald Kivumbi Muganga, Adolphus Cherop.

**Funding acquisition:** Joan Rokani Bayowa.

**Investigation:** Joan Rokani Bayowa, Joan N. Kalyango, Joseph Baruch Baluku, Richard Katuramu, Emmanuel Ssendikwanawa, Jane Frances Zalwango, Adolphus Cherop.

**Methodology:** Joan Rokani Bayowa, Joan N. Kalyango, Joseph Baruch Baluku, Richard Katuramu, Emmanuel Ssendikwanawa, Jane Frances Zalwango, Rebecca Akunzirwe, Stella Maris Nanyonga, Judith Ssemasazi Amutuhaire, Ronald Kivumbi Muganga, Adolphus Cherop.

**Project administration:** Joan Rokani Bayowa.

**Resources:** Joan Rokani Bayowa.

**Supervision:** Joan Rokani Bayowa, Joan N. Kalyango, Joseph Baruch Baluku, Richard Katuramu.

**Validation:** Joan Rokani Bayowa, Joan N. Kalyango, Joseph Baruch Baluku, Emmanuel Ssendikwanawa.

**Visualization:** Joan Rokani Bayowa, Joan N. Kalyango.

**Writing – original draft:** Joan Rokani Bayowa, Emmanuel Ssendikwanawa, Jane Frances Zalwango, Rebecca Akunzirwe, Stella Maris Nanyonga, Judith Ssemasazi Amutuhaire, Ronald Kivumbi Muganga, Adolphus Cherop.

**Writing – review & editing:** Joan Rokani Bayowa, Joan N. Kalyango, Joseph Baruch Baluku, Jane Frances Zalwango.

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
