## [Decision Letter · Decision Letter 0]

3 Oct 2022

PGPH-D-22-01063

Mortality rate and associated factors among  patients co-infected with drug resistant Tuberculosis/HIV in Mulago National Referral Hospital, Uganda, a retrospective cohort study

Dear Dr. %Bayowa%,

Thank you for submitting your manuscript to PLOS Global Public Health. After careful consideration, we feel that it has merit but does not fully meet PLOS Global Public Health’s publication criteria as it currently stands. Therefore, we invite you to submit a revised version of the manuscript that addresses the points raised during the review process.

We look forward to receiving your revised manuscript.

Kind regards,

Zulma Vanessa Rueda, M.D. Ph.D.

Academic Editor

Journal Requirements:

1. We noticed that you entered only a single author in the Editorial Manager submission form, yet you listed multiple authors on the manuscript file, and you indicated their contributions in the "Opposed Reviewer" box. Please ensure that all authors' details are entered correctly and completely in Editorial Manager. This should be done at the time of submission so that the journal can complete our internal checks and so that all co-authors receive relevant communications about the submission. If you have any questions, please email the journal office.

2. Please provide your detailed Financial Disclosure statement. This is published with the article. It must therefore be completed in full sentences and contain the exact wording you wish to be published.

3. We ask that a manuscript source file is provided at Revision. Please upload your manuscript file as a .doc, .docx, .rtf or .tex.

Mortality rate and associated factors among patients co-infected with drug resistant tuberculosis and HIV at Mulago National Referral Hospital; a retrospective cohort study.

4. Please provide separate figure files in .tif or .eps format only and remove any figures embedded in your manuscript file. Please also ensure that all files are under our size limit of 10MB.

5. Please provide a complete Data Availability Statement in the submission form, ensuring you include all necessary access information or a reason for why you are unable to make your data freely accessible. If your research concerns only data provided within your submission, please write "All data are in the manuscript and/or supporting information files" as your Data Availability Statement.

Additional Editor Comments (if provided):

Dear authors,

Thanks for submitting your paper. I agree with the reviewer's comments regarding your manuscript. I invite you to review and address the reviewer's comments.

Reviewers' comments:

Reviewer's Responses to Questions

**Comments to the Author**

1. Does this manuscript meet PLOS Global Public Health’s publication criteria? Is the manuscript technically sound, and do the data support the conclusions? The manuscript must describe methodologically and ethically rigorous research with conclusions that are appropriately drawn based on the data presented.

Reviewer #1: Partly

Reviewer #2: Partly

2. Has the statistical analysis been performed appropriately and rigorously?

Reviewer #1: No

Reviewer #2: No

3. Have the authors made all data underlying the findings in their manuscript fully available (please refer to the Data Availability Statement at the start of the manuscript PDF file)?

Reviewer #1: No

Reviewer #2: Yes

4. Is the manuscript presented in an intelligible fashion and written in standard English?

Reviewer #1: No

Reviewer #2: No

5. Review Comments to the Author

Reviewer #1: This paper discusses high mortality in a relatively young DR-TB/HIV co-infected cohort from Uganda. It will indeed contribute to the increasing pool of literature on DR-TB and HIV in East Africa. The manuscript is well-written with a good presentation. To make this paper more interesting and informative, the authors should, however, take into account the following issues.

Major comments:

1. Considering that this is a cohort study examining the mortality rate, I am curious as to why the authors chose modified poisson methods to determine the rates rather than calculating incidence rates (per person years). I would advise a survival analysis and Cox regression models, unless there is a compelling reason not to.

2. The descriptions of study participants and inclusion criteria are ambiguous. Does the study solely include DR-TB/HIV patients taking second line TB drugs? OR were participants just newly diagnosed with DR-TB?

3. It would be very helpful and in accordance with the STROBE standard to create a study flow diagram.

4. If possible, I would like the authors to provide the following significant DR-TB and HIV infection characteristics:

a. Drug resistant profiles, including mono, poly, MDR, XDR, Rifampicin or otherwise.

b. Type of TB (pulmonary, extra-pulmonary, or both)

c. Location of TB

d. Diagnosis order (TB first, HIV first, or diagnosed at the same time), time between HIV and TB infection

e. TB medications specifically (moxifloxacin, levofloxacin, bedaquiline, or linezolid: they are associated with significantly decreased odds of death.)

5. A nice meta-analysis and some sub-analyses on MDR-TB and HIV mortality were provided by Bisson et al. in 2020 (https://doi.org/10.1016/S0140-6736(20)31316-7), which may guide further revision of the analyses in this manuscript.

Minor comments:

1. MUAC cut-off point appears to be poorly correlated with BMI categories, suggesting a severe illness or acute malnutrition. I would suggest lowering the cut-off point to roughly 18.5, which some say is appropriate for all settings, to accurately reflect this position. (Global MUAC cut-offs for adults: Technical Consultation: http://pdf.usaid.gov/pdf_docs/PA00T7K9.pdf)

2. It doesn't seem necessary to include Figure 2. Include it as text in the Results section.

3. Table 2 and 3 should be presented in one table.

Reviewer #2: The authors retrospectively reviewed data of 390 patients with a DR-TB/HIV co-infection and determined the mortality rate and factors associated with mortality among those at Mulago Hospital.

There are some minor and significant suggestions I think could improve the document:

Major:

• Were there any changes in the mortality rate during the years of the study? This information could help show any trend.

• Other variables can impact the mortality outcome, for example, CD4 count, viral load, localization of TB (pulmonary or extrapulmonary TB), etc. Could you let me know if you collected this information? If not, Why not?

• Is the median time to mortality reported? E.g. time since diagnosis or time since treatment initiation

• Please explain why you decided to divide the age into two categories.

• How many patients were included in the final model? Please clarify.

• The authors mention they checked interaction among the variables; please specify what interaction you checked and what are the results.

• You used the BMI and MUAC in the same model. Are those anthropometric measures related? Did you evaluate the collinearity?

• You mention the study was carried out in the Mulago Hospital tuberculosis unit. Having access to the patient files, why did you not check for other comorbidities that could be associated with mortality, like diabetes? If you did, I think you should mention it and include this information in the paper. If not, why not?

• The authors review the mortality rates for DR-TB/HIV reported by other publications showing lower mortality rates in coinfected patients. Given this, I suggest that the authors discuss in depth the implications that these high rates represent for the country and issue some recommendations.

• Were there any changes in the TB treatment administration during the study? In addition, the authors mention, "This is because, by the time this study was undertaken, antiretroviral treatment (ART) was rolled out countrywide and was available in Uganda". Was this change during the first year of the study?

Minor:

• Please check the spelling and spaces used in the document, including the tables.

• Could you update the references? For example, reference #1 is the WHO 2018 TB report; there is a new version (2021).

• Please, check the results section regarding the date that the patients were admitted. There is a mistake there.

• Please, explain what MUAC means the first time it appears.

• Review the structure of the references; some of them are incomplete

• There are two references in the results (Mortality rate). What do those mean?

6. PLOS authors have the option to publish the peer review history of their article (what does this mean?). If published, this will include your full peer review and any attached files.

**Do you want your identity to be public for this peer review?** For information about this choice, including consent withdrawal, please see our Privacy Policy.

Reviewer #1: No

Reviewer #2: **Yes: **Mariana Herrera Diaz

---

## [Decision Letter · Decision Letter 1]

17 Apr 2023

PGPH-D-22-01063R1

Mortality rate and associated factors among  patients co-infected with drug resistant Tuberculosis/HIV in Mulago National Referral Hospital, Uganda, a retrospective cohort study

Dear Dr. Bayowa,

Thank you for submitting your manuscript to PLOS Global Public Health. After careful consideration, we feel that it has merit but does not fully meet PLOS Global Public Health’s publication criteria as it currently stands. Therefore, we invite you to submit a revised version of the manuscript that addresses the points raised during the review process.

We look forward to receiving your revised manuscript.

Kind regards,

Zulma Vanessa Rueda, M.D. Ph.D.

Academic Editor

Journal Requirements:

2. Please ensure that the Title in your manuscript file and the Title provided in your online submission form are the same.

Additional Editor Comments (if provided):

Dear authors,

Thanks for your work and time to address reviewer's comments. Please see and address the minor comments of one of the reviewers.

In addition, I encourage you to incorporate within the manuscript some of the answers that you provided to the reviewers in the previous round of reviews.

I also encourage you to review and edit accordingly the manuscript as there are no spaces between words and punctuation signs in some section of the paper.

Finally, one of the reviewer's comments in the previous and current round of reviews, mentioned that you are using references from other studies within the results section. The result section should focus only on your research results, rather than other published papers. Please verify and correct accordingly. You can compare your results and elaborate your owns figures comparing with previous literature, in the discussion section.

Thanks,

Reviewers' comments:

Reviewer's Responses to Questions

**Comments to the Author**

1. If the authors have adequately addressed your comments raised in a previous round of review and you feel that this manuscript is now acceptable for publication, you may indicate that here to bypass the “Comments to the Author” section, enter your conflict of interest statement in the “Confidential to Editor” section, and submit your "Accept" recommendation.

Reviewer #2: All comments have been addressed

Reviewer #3: All comments have been addressed

2. Does this manuscript meet PLOS Global Public Health’s publication criteria? Is the manuscript technically sound, and do the data support the conclusions? The manuscript must describe methodologically and ethically rigorous research with conclusions that are appropriately drawn based on the data presented.

Reviewer #2: Yes

Reviewer #3: Yes

3. Has the statistical analysis been performed appropriately and rigorously?

Reviewer #2: Yes

Reviewer #3: Yes

4. Have the authors made all data underlying the findings in their manuscript fully available (please refer to the Data Availability Statement at the start of the manuscript PDF file)?

Reviewer #2: Yes

Reviewer #3: No

5. Is the manuscript presented in an intelligible fashion and written in standard English?

Reviewer #2: Yes

Reviewer #3: Yes

6. Review Comments to the Author

Reviewer #2: The authors did a good job answering each of the points, however some of the answers are not apparent in the paper. I recommend the authors to review and include them in the final version.

Reviewer #3: This is a very interesting work, which presents the relationship between the characteristics of patients with drug resistant TB/HIV coinfection and mortality. There are some recommendations that can make some improvements in this manuscript form.

In the abstract ART is not explained at first mentioned, neither MUAC.

There are some spaces missing between some words in the abstract and the text body.

Table 1: in the footnote to the table there are some abbreviations missing.

Results: there are some data that goes better in the discussion section, as those are not results from this study, for example in mortality rate “This is a consistent with a Ugandan study by Walusimbi et al, that registered high deaths in 2016[16].This trend could be explained by provider and health system barriers, including lack of knowledge, skills and negative attitude on TB among health providers, stockout of supplies such as medicines, high staff turnover, inability to track and follow-up TB patients, and poor service coordination[17]. Additionally, comorbidities with TB have also been found to a major risk for TB mortality [18][19].]”

Format reference need to be reviewed.

7. PLOS authors have the option to publish the peer review history of their article (what does this mean?). If published, this will include your full peer review and any attached files.

**Do you want your identity to be public for this peer review?** For information about this choice, including consent withdrawal, please see our Privacy Policy.

Reviewer #2: **Yes: **Mariana Herrera Diaz

Reviewer #3: No

---

## [Editor Report · Decision Letter 2]

8 Jun 2023

Mortality rate and associated factors among  patients co-infected with drug resistant Tuberculosis/HIV in Mulago National Referral Hospital, Uganda, a retrospective cohort study

PGPH-D-22-01063R2

Dear %Dr. Bayowa%,

We are pleased to inform you that your manuscript 'Mortality rate and associated factors among  patients co-infected with drug resistant Tuberculosis/HIV in Mulago National Referral Hospital, Uganda, a retrospective cohort study' has been provisionally accepted for publication in PLOS Global Public Health.

Best regards,

Zulma Vanessa Rueda, M.D. Ph.D.

Academic Editor

Thanks for your work and contribution of this paper.

The authors addressed the reviewer's comments. My only suggestion during the editing it that there are some words that have no space, for example, line 364, "TB/HIV coinfectedpatients".

I also encourage authors to use person-centered language in the final editing of the paper. This guide from STOP TB Partnership is a wonderful tool for all of us, as researchers and community in general, to learn and use a person-centered language. The link where authors can find the guide is: https://www.stoptb.org/words-matter-language-guide

Thanks to the journal and authors for the opportunity to edit this important paper.